# Transdermal Permeation Assays of Curcumin Aided by CAGE-IL: In Vivo Control of Psoriasis

**DOI:** 10.3390/pharmaceutics14040779

**Published:** 2022-04-02

**Authors:** Rodrigo Boscariol, Érika A. Caetano, Denise Grotto, Raquel M. Rosa-Castro, José M. Oliveira Junior, Marta M. D. C. Vila, Victor M. Balcão

**Affiliations:** 1Phagelab, Laboratory of Biofilms and Bacteriophages, University of Sorocaba (UNISO), Sorocaba 18023-000, Brazil; rodrigoboscariol@yahoo.com.br (R.B.); 00075448@aluno.uniso.br (É.A.C.); denise.grotto@prof.uniso.br (D.G.); raquel.rosa@prof.uniso.br (R.M.R.-C.); jose.oliveira@prof.uniso.br (J.M.O.J.); 2Department of Biology and CESAM, University of Aveiro, Campus Universitário de Santiago, P-3810-193 Aveiro, Portugal

**Keywords:** choline and geranic acid ionic liquid, transdermal permeation, curcumin, psoriasis, in vivo assays

## Abstract

Psoriasis is a clinically heterogeneous skin disease with an important genetic component, whose pathophysiology is not yet fully understood and for which there is still no cure. Hence, alternative therapies have been evaluated, using plant species such as turmeric (*Curcuma longa* Linn.) in topical preparations. However, the stratum corneum is a barrier to be overcome, and ionic liquids have emerged as potential substances that promote skin permeation. Thus, the main objective of this research was to evaluate a biopolysaccharide hydrogel formulation integrating curcumin with choline and geranic acid ionic liquid (CAGE-IL) as a facilitator of skin transdermal permeation, in the treatment of chemically induced psoriasis in mice. The developed gel containing curcumin and CAGE-IL showed a high potential for applications in the treatment of psoriasis, reversing the histological manifestations of psoriasis to a state very close to that of normal skin.

## 1. Introduction

Psoriasis is a relatively common skin disease, being, from a pathological point of view, chronic dermatitis with rapid uncontrolled proliferation of epithelial cells on the surface, hyperemia and dense lymphocytic infiltration [1]. Essentially, the disease can start at any stage of life and persist for a long time, with permanent or periodic eruptions [2,3]. It is recognized as a genetically modulated disease [4] that results in a chronic, autoimmune, standard inflammatory condition [1] characterized by thick, red, scaly patches or plaques distributed throughout the body. The condition is usually governed by components of the patient’s immune system [5,6]. A variety of topical and systemic drug therapies exist, and treatment regimens must be optimized for achieving optimal compliance and benefit [7,8,9]. However, despite the availability of different topical agents and systemic therapeutic options, none of the treatments provide excellent clinical results without the risk of side effects [10,11].

More recently, studies point to curcumin (CUR) as a potential asset in the therapy of psoriasis [12,13,14,15,16]. However, curcumin is practically insoluble in water and has low availability, so for use in topical preparations it is necessary to use substances that favor its permeation. Transdermal permeation aims to transport active pharmaceutical ingredients (APIs) in the skin tissue to the deeper layers, with clear therapeutic advantages compared to giving APIs orally or parenterally [17,18]. However, the stratum corneum is a natural protective barrier, with a low permeation rate. Thus, the development of permeation-promoting systems that promote a transient change in the structure of the skin, enabling more efficient bioaccessibility and permeability of bioactive molecules, is of extreme importance [14,19,20]. Ionic liquids have emerged as facilitators of skin permeation for protein entities of interest [19,20,21]. Some ionic liquids such as choline and geranic acid ionic liquid (CAGE-IL) have the ability to cause transient alterations in the intercellular and intracellular lipids and the protein organization of the stratum corneum, allowing the penetration of bioactive molecules carried by these liquids [19,20,22,23]. In addition, ionic liquids can be incorporated into drug delivery systems, among which we can highlight hydrogels [24].

For psoriasis research trials, there are several categories of experimental animal models that can be used, among which the “direct induction” type is most viable for drug development and screening. Imiquimod (IMQ) is a ligand for Toll-like 7/8 receptors that activates macrophages, monocytes and dendritic cells and, when administered directly to the skin of mice, induces skin damage similar to psoriasis [25,26]. The agonist effect of imiquimod on the Toll-like receptor causes dermal dendritic cells to express cytokines such as interferon-α (IFN-α), tumor necrosis factor-α (TNF-α) and the interleukins IL-1, IL-6 and IL-8 [27]. This innate immune response can lead to adverse events such as redness, swelling and a burning sensation [28,29]. The IMQ-treated mouse model is the most widely explored animal model for studying psoriasis in vivo. It is a rapid and convenient model, which allows the elucidation of the mechanisms underlying the disease and the evaluation of new therapies against psoriasis [30]. In general, three days after application of imiquimod, mouse skin begins to resemble human skin with psoriasis, in terms of erythema and skin thickening, skin desquamation, epidermal changes (acanthosis, parakeratosis) and neoangiogenesis, as well as showing an inflammatory infiltrate consisting of T cells, neutrophils and dendritic cells [29,31]. However, there are differences in both the mechanism and the clinical features between imiquimod-induced psoriasis in mice and human psoriasis. In view of its acute and severe characteristics in the progression of the disease, the use of this animal model is restricted to the development of preclinical medications [29,32]. To treat the lesions developed by the experimental model, low-cost topical treatments based on natural products such as curcumin have attracted scientific attention [14,19,33,34].

Ionic liquids can transiently disrupt the skin barrier function by modifying the disposition of the corneocytes of the stratum corneum [19,33,35]. The biological outcomes of ionic liquids, including transdermal delivery, antibacterial activity and cytotoxicity are fully related to the chemical properties and their constituent cations and anions [35,36]. Compared to traditional organic solvents with small alkyl chains, such as ethanol, they are less toxic to cells, thus minimizing problems associated with solvent-induced skin irritation [37]. In particular, the ionic liquid of choline and geranic acid (CAGE-IL) has been successfully used by our research group to increase the skin permeation of several substances such as insulin [20], bacteriophage particles [24,38] and curcumin [19,33]. It has also been used by other research groups [39]. Therefore, CAGE-IL was chosen as a facilitator of dermal permeation in this study.

The objective of this research was to develop and evaluate in vivo a locust bean gum (LBG) gel integrating CAGE-IL, aiming at the transdermal permeation of curcumin for the treatment of imiquimod-induced psoriasis using a mouse model.

## 2. Materials and Methods

### 2.1. Materials

#### 2.1.1. Test Animals

A total of 40 mice of the BALB/c strain (20 males and 20 females), aged 7–9 weeks, with average weights of ca. 25 g, were acquired from the Anilab bioterium (Paulínia/SP, Brazil) and used in the in vivo assays. All animals were maintained in the bioterium of the University of Sorocaba (Sorocaba/SP, Brazil). All procedures were performed in the vivarium of the University of Sorocaba, where the animals were kept in cages with a maximum number of 2 to 3 animals, grouped by gender, kept under constant temperature conditions of 24 ± 1 °C, with a night–day cycle of 12:12 h and relative air humidity between 60 and 70%. Wood shavings were placed in the cages, and the animals were fed standard rodent chow and water ad libitum. All animals underwent an acclimatization period of 7 d prior to the beginning of the experiments. The acquisition of the animals and the experimental procedures performed were approved by the Ethics Committee on the Use of Animals of the University of Sorocaba (CEUA-UNISO), with approval resolution CEUA-UNISO no. 199/2021.

#### 2.1.2. Chemicals

Water (ultra)purification was performed in a Master System All (model MS2000, Gehaka, São Paulo, Brazil). Geranic acid (85% stabilized; cat.# W412101-1KG-K), choline bicarbonate (cat.# C7519-500ML), locust bean gum (LGB) (cat.# No. G0753) and curcumin (CUR) (cat.# No. C1386) were acquired from Sigma-Aldrich (St. Louis, MO, USA). HPLC-grade methanol (LiChrosolv^®^, CAS No: 67-56-1) was purchased from Merck (Darmstadt, Germany). Imiquimod cream (50 mg/g) (Modik^®^) was acquired from Germed Farmacêutica LTDA. (Hortolândia/SP, Brazil).

### 2.2. Experimental Procedures

#### 2.2.1. Synthesis of Choline and Geranic Acid Ionic Liquid (CAGE-IL)

Choline and Geranic acid Ionic Liquid (CAGE-IL) was obtained in accordance with the procedures described in [20,22,36]. In brief, 48 mL of geranic acid (CAS No. 459-80-3; Sigma-Aldrich, St. Louis, MO, USA), 20 mL of choline bicarbonate at 80% (*w/v*) (CAS No. 62-49-7; Sigma-Aldrich, St. Louis, MO, USA) and 20 mL of methanol (CAS No. 67-56-1; Chemco Indústria e Comércio Ltda., Hortolândia/SP, Brazil) were added to a 1000 mL round-bottom flask and mixed until CO_2_ production ceased. The solvent was removed in a rotary evaporator (BÜCHI Labortechnik GmbH, model R-215, Essen, Germany) at 60 °C, for ca. 20 min. The prepared CAGE-IL was transferred to a 50 mL Falcon tube and nitrogen bubbled to scavenge any dissolved oxygen, thereby preventing oxidation. The tube was capped and sealed with Parafilm™ (Bemis Flexible Packaging, Neenah, WI, USA).

#### 2.2.2. Preparation and Characterization of Biopolysaccharide Gels

***Biopolysaccharide gel with curcumin*****.** Gel formulations integrating curcumin were prepared using locust bean gum (LBG) (Ref. No. G0753; Sigma-Aldrich, St. Louis, MO, USA) as a gelling agent. For each gel, precise amounts of curcumin (CUR) (Ref. No. C1386; Sigma-Aldrich, St. Louis, MO, USA) with 95% purity (2.087%, *w/w*), methylparaben (0.1%, *w/w*) and LBG (2%, *w/w*) were dispersed in ultrapure water and mixed under continuous magnetic stirring.

***Biopolysaccharide gel with curcumin and CAGE-IL*.** Precise mass concentrations of curcumin (CUR) (Ref. No. C1386; Sigma-Aldrich, St. Louis MO, USA) with 95% purity (2.087%, *w/w*), methylparaben (0.1%, *w/w*) and locust bean gum (LBG) (2%, *w/w*) (Ref. No. G0753; Sigma-Aldrich, St. Louis, MO, USA) were dispersed in ultrapure water under continuous magnetic stirring. CAGE-IL was then added up to a final concentration of 2% (*w/w*). In the preparation of the gels, careful homogenization was performed to avoid air incorporation.

The percentage of curcumin utilized (2%, *w/w*) was established based on the work of Patel et al. [39], and the most adequate percentage of CAGE-IL (2%, *w/w*) was selected on the basis of previous studies [19,33]. The percentage of LGB used was experimentally determined by testing weight percentages from 2% to 3%, as indicated by its gelling capacity [23,40], with the percentage of 2% (*w/w*) being determined as the most suitable for the intended purpose. Table 1 displays the details of all gel formulations prepared.

Full physicochemical characterization of the gel formulations formed the subject of a previous study [33], and in this study, thermal characterization was additionally performed in order to study the influence of the various components of the gel formulations on thermal stability.

***Thermal characterization of biopolysaccharide gels* via *differential scanning calorimetry (DSC) analyses*.** Thermal characterization of samples of pure curcumin (CUR); CAGE-IL; gel containing 2% (*w/w*) LBG and 2% (*w/w*) CUR; gel containing 2% (*w/w*) LBG, 2% (*w/w*) CUR and 2% (*w/w*) CAGE-IL; and gel containing 2% (*w/w*) LBG, 2.087% (*w/w*) commercial turmeric (containing 2% (*w/w*) CUR) and 2% (*w/w*) CAGE-IL, was accomplished via differential scanning calorimetry (DSC) analyses in a microcalorimeter from Shimadzu (model DSC-60, Kyoto, Japan) coupled with a TA 60W thermal analyzer also from Shimadzu, following the procedure described in detail in [20,41], using sample weights from ca. 1.24 mg to ca. 2.55 mg.

#### 2.2.3. Experimental Design and Induction of Psoriasis

After an acclimatization period of one week, the mice were randomly divided into eight groups: G1, G2, G3, G4, G5, G6, G7 and G8, and placed in individual cages, with five animals (10 ears, n = 10) per group. In group G1 (3 males + 2 females), there was no application, characterizing this group as the absolute control group; group G2 (2 males + 3 females) received LBG gel (2%, *w/w*) in the ears for 20 consecutive days; group G3 (3 males + 2 females) received LBG gel (2%, *w/w*) with CUR (2%, *w/w*) in the ears for 20 consecutive days; group G4 (2 males + 3 females) received LBG gel (2%, *w/w*) with CUR (2%, *w/w*) and CAGE-IL (2%, *w/w*) in the ears for 20 consecutive days. Applications in groups G2 to G4 occurred in the right ear, keeping the left ear as a negative control. In groups G5 to G8, all animals received imiquimod (IMQ) (commercially available in the form of a 5% (*w/w*) cream) at a dose of 25 mg/day, which corresponded to 1.25 mg of the active component, on the inner surface of both ears once a day for 10 consecutive days [42]. After this period, the G5 group (2 males + 3 females) did not receive any further treatment, characterizing this group as the IMQ control group. Group G6 (2 males + 3 females) received treatment with LBG gel (2%, *w/w*); group G7 (3 males + 3 females) received treatment with LBG gel (2%, *w/w*) with CUR (2%, *w/w*) at a dose of 0.5 mg/day; and the G8 group (3 males + 2 females) received treatment with LBG gel (2%, *w/w*) with CUR (2 %, *w/w*) and CAGE-IL (2%, *w/w*) at a dose of 0.5 mg/day [43]. Groups G6 to G8 received treatments in both ears.

Mice received topical IMQ cream on the epidermis on the dorsal and ventral surfaces once a day for 10 consecutive days. The IMQ cream was chosen because of its direct action, as it can be applied directly to the animal’s epidermis. In general, mouse skin begins to resemble human skin with psoriasis within three days of application of IMQ [25,29].

#### 2.2.4. Assessment of Inflammation Severity

The procedure for monitoring and recording the severity of the inflammation of the skin of the animals’ ears was performed using a scoring system based on the Psoriasis Area Severity Index (PASI) [27,44,45]. Observations of erythema and peeling of the skin were independently recorded by assigning values on a scale of 0 to 4 as follows: 0, none; 1, light; 2, moderate; 3, intense; 4, very intense. The evolution of the thickness of the ears was also monitored using a pachymeter (Digimess Instrumentos de Precisão, Ltda—São Paulo/SP, Brazil) throughout all the experiments, performing measurements in triplicate in all groups of animals.

#### 2.2.5. Mouse Euthanasia

All mice were euthanized 28 days after the start of treatment(s), which was a period of time sufficient for the appearance of inflammatory lesions and scales in the animals treated with IMQ [46]. The animals were euthanized by applying ketamine (100 mg/kg) associated with xylasin (900 mg/kg) [47], after which the ears were surgically removed for histological analysis.

#### 2.2.6. Preparation of Histological Cuts

Histological analysis of the skin of the mice ears was performed after fixation. The ears were placed in buffered formalin for 8 h. Then, the samples were washed with ultrapure water and the tissues stored in 70% (*v/v*) ethanol. Afterwards, the tissues were treated in an automatic tissue processor (O Patologista^®^, model PT12, Guarulhos/SP, Brazil), and the pieces were fixed in paraffin in the sequence [27,45]: (i) dehydration (passage through 70% (*v/v*) ethanol; 50 min in 90% (*v/v*) ethanol; 150 min in 100% (*v/v*) ethanol), (ii) clarification with xylene for 120 min and (iii) paraffin impregnation for 120 min. The histological sections were prepared in a manual rotary microtome (O Patologista^®^, model MR 2014, Guarulhos/SP, Brazil) and were subsequently stained with hematoxylin and eosin ((i) dewaxing with xylene for 30 min, (ii) hydration with 100% (*v/v*) ethanol for 20 s, (iii) hydration with 95% (*v/v*) ethanol) for 10 s, (iv) hydration with 70% (*v/v*) ethanol for 10 s, (v) washing with ultrapure water for 2 s, (vi) staining with hematoxylin for 10 min, (vii) washing with running water for 3 min, (viii) staining with eosin for 3 min, (ix) washing with ultrapure water for 2s, (x) dehydration with 70% (*v/v*) ethanol for 10 min, (xi) dehydration with ethanol at 100% (*v/v*) for 20 min, (xii) dye fixation and material preservation with xylene for 1s, (xiii) dye fixation and preservation of material with xylene until the slide was mounted and finally (xiv) assembly with histological resin) [48,49,50]. The analysis of the histological sections was performed using an optical microscope with a digital camera (biological microscope, model Axio Lab.A1, Zeiss^®^, Oberkochen, Germany).

#### 2.2.7. Statistical Analyses

The experimental results were subjected to extensive descriptive statistical analysis using the Statistica™ software (version 10, StatSoft Inc., Hamburg, Germany). Data variability was evaluated using the ANOVA test: a non-parametric test used to compare three or more independent samples. A significance level of 0.05 was adopted for the mean and standard deviation. Having rejected the null hypothesis (*p* < 0.05), indicating a difference between at least two groups of animals, the Tukey post hoc test was applied (*p* < 0.001). For skin thickness data, a one-way ANOVA test was performed followed by Tukey’s test. For the score data (erythema and desquamation), Kruskal–Wallis and Tukey tests were performed.

## 3. Results

The choline and geranic acid ionic liquid (CAGE-IL) was synthesized according to the procedures described in detail in [19,20,33], integrated as permeation enhancer in a locust bean gum (LBG) biopolysaccharide gel containing curcumin (CUR) and used in the in vivo transdermal permeation assays for controlling imiquimod (IMQ)-induced psoriasis in mice.

### 3.1. Thermal Characterization of Biopolysaccharide Gels via Differential Scanning Calorimetry (DSC) Analyses

The DSC thermograms of samples of CAGE-IL, pure CUR, commercial turmeric, LBG gel formulation integrating pure CUR, LBG gel formulation integrating pure CUR and CAGE-IL and LBG gel formulation integrating commercial turmeric and CAGE-IL are displayed in Figure 1.

No significant differences were observed (see Figure 1), as the endothermic events displayed by the gel formulations integrating CUR (blue line in Figure 1) or CUR and CAGE-IL (red line in Figure 1) were quite similar, with the addition of CAGE-IL to the gel formulation having the effect of displacing the endothermic peak from 92.33 °C to 80.64 °C, with a concomitant decrease in the melting enthalpy from 832.37 J/g to 584.99 J/g (Figure 1).

### 3.2. Macroscopic/Microscopic Analysis of Mouse Ears

During the in vivo experiments, macroscopic changes were observed in the lesions formed by the application of IMQ cream on the epidermis of the mouse ears. Normal/docile behavior of the animals was noted due to the fact that the minimal manipulation necessary for the application of the cream did not generate any type of stress to the animal. After applying the treatments proposed here, it was possible to observe macroscopically the regression of lesions caused by IMQ in the epidermis of the animals’ ears, with a more evident effect observed after treatment with the gel containing CUR (2%, *w/w*) and CAGE-IL (2%, *w/w*), classifying the topical treatment as the gold standard for lesions in the epidermis. No significant weight changes were observed in the animals during the experiments, and no animals were lost.

#### 3.2.1. Control Groups

The mice in the control group (G1) did not receive any type of application or treatment during the experiments. For 20 days, mice from groups G2, G3 and G4 received the application of LBG gel (2%, *w/w*), LBG gel (2%, *w/w*) with CUR (2%, *w/w*) and LBG gel (2%, *w/w*) with CUR (2%, *w/w*) and CAGE-IL (2%, *w/w*), respectively. In these groups, the formation of erythema and the desquamation of the skin of the ears were observed. The monitoring of the evolution of the thickness of the ears also indicated no change during the entire period of the experiments, in either male rats (Figure 2) or female rats (Figure 3), demonstrating the safety of the use of the formulations.

The choline and geranic acid ionic liquid (CAGE-IL) in the LBG gel formulation (2%, *w/w*) containing CUR (2%, *w/w*) applied to the G4 group promoted a very noticeable deposition of CUR in the ears of both male and female mice (Figure 2e,f and Figure 3e,f), which was not observed in the animals of the G3 group, for which the LBG gel formulation (2%, *w/w*) applied did not contain CAGE-IL (Figure 2c,d and Figure 3c,d).

Figure 4 displays the evolution of the thickness of the ears of the animals in the control groups, over 20 d of application of LBG gel (2%, *w/w*), LBG gel (2%, *w/w*) with CUR (2%, *w/w*) and LBG gel (2%, *w/w*) with CUR (2%, *w/w*) and CAGE-IL (2%, *w/w*).

#### 3.2.2. Induced Psoriasis Groups

The animals in groups G5, G6, G7 and G8 were subjected to the psoriasis induction model via application of IMQ over 10 d, with the aim of developing symptoms similar to those of human psoriasis. After this period of time, groups G6, G7 and G8 received treatment in the form of a gel formulation for 10 consecutive days, once a day, in both ears, with the animals of the G6 group being treated with LBG gel (2%, *w/w*), the animals of the G7 group treated with LBG gel (2%, *w/w*) containing CUR (2%, *w/w*) and the animals of the G8 group treated with LBG gel (2%, *w/w*) containing CUR (2%, *w/w*) and CAGE-IL (2%, *w/w*). The animals in the G5 group did not receive any treatment, characterizing this group as the control group for IMQ-induced psoriasis, where a gradual remission of the lesions caused by the natural process of the mouse body’s response was observed. The application of the IMQ cream produced the characteristics of psoriasis inflammation, erythema and scaling of the skin in all groups of animals, both male and female, as expected. The manifestations of psoriatic lesions and the results of the various treatments applied to the epidermis of the ears of the mice can be visualized in Figure 5 and Figure 6 for male and female mice, respectively.

### 3.3. Treatment of Psoriasis

To measure the severity of inflammation of the epidermis of the ear, the Psoriasis Area Severity Index (PASI) scoring system was used, with erythema and scaling of the skin being independently rated on a scale of 0 to 4. Figure 7 presents the results obtained for the evolution of erythema and skin desquamation induced by imiquimod (IMQ) in male and female mice.

Figure 8 presents the results obtained for the evolution of erythema and ear skin peeling induced by imiquimod (IMQ) and the comparison of the remission obtained via application of the treatments with the LBG gel formulations.

### 3.4. Assessment of Mouse Ear Epidermis Thickness

Control groups for the formulated gels were exposed to treatments for 20 consecutive days in order to verify the safety of using the developed formulations. For the groups of animals with induced psoriasis, the animals (males and females) were exposed to IMQ for 10 consecutive days, after which the animals were treated with the formulated gels for another 10 consecutive days, with only the control group being exposed to IMQ without the application of any treatment (following the natural course of the mouse organism). The results for the increase in the thickness of the epidermis of the skin of the mouse ears and the remission after the applied treatments are displayed in Figure 9a for the male mice and in Figure 9b for the female mice, for a test period of 20 days. All ear thickness measurements were performed using a pachymeter (Digimess Instrumentos de Precisão Ltd., São Paulo/SP, Brazil).

## 4. Discussion

The gel formulations produced basically consisted of water and locust bean gum (LBG). LBG was dissolved in water and heated at 70 °C for 2 h for complete dissolution [51]. CUR, as a hydrophobic molecule with low solubility in water (particularly in an acidic or neutral environment, where it remains fully protonated), must be dissolved in some solvent or hydrophobic substance before its incorporation into an aqueous matrix [52]. Thus, the CAGE-IL, in addition to enhancing permeation, aided in increasing CUR dissolution. The percentage chosen for the ionic liquid was established as the most adequate in previous studies [19,33].

Among the various topical pharmaceutical forms available, gels and hydrogels are very popular and have high acceptance by patients, due to their ease of application. Gels based on natural polysaccharides can be produced without the presence of toxic substances, which makes them interesting for the development of delivery systems for bioactive molecules [53]. A biopolysaccharide that has attracted attention is LBG, a plant galactomannan extracted from locust bean seeds (from the carob tree, *Ceratonia siliqua*) [54,55]. This polymer has a series of attractive characteristics for biopharmaceutical applications, including its high gelling capacity, biodegradability, low toxicity and low cost [55]. Due to these characteristics, it was chosen as the main component in the gel formulation.

The gel developed presented suitable characteristics for dermal applications, as pointed out in a previous study [33], and DSC analysis was carried out in order to evaluate the thermal stability of the components integrated into the gel formulation. DSC is a thermal analysis method for measuring how the physical properties of a sample change with temperature over time. During a linear increase in temperature, DSC measures the amount of heat that is liberated or absorbed by the sample on the basis of a temperature difference between the sample and a reference material [56]. Hence, the three formulations tested were studied via differential calorimetry to verify whether any of the added component(s) could change the thermal stability, causing a loss of stability of the gel formulations. No significant differences were observed (see Figure 1), as the endothermic events displayed by the gel formulations integrating CUR (blue line in Figure 1) or CUR and CAGE-IL (red line in Figure 1) were quite similar, with the addition of CAGE-IL to the gel formulation having the effect of a displacing the endothermic peak towards a lower temperature, with a concomitant decrease in the melting enthalpy of 29.7% (Figure 1). Thus, the thermal stabilities of the gel formulations with CUR, with or without CAGE-IL, were similar.

One can observe very similar thermal events for the three gel formulations, and the ratios of the melting enthalpies of the gel without CAGE-IL and the gels with CAGE-IL were of the same order of magnitude (1.42× and 1.10×, respectively, for the LBG gels integrating pure CUR and commercial turmeric). Integrating CAGE-IL into the LBG formulations promoted a displacement in the major endothermal event from 92.33 °C to ca. 81–82 °C (Figure 1), thereby improving the plasticity of the LBG gel formulation at lower temperatures. The major endothermic event in pure CUR was the sharp peak at 179.55 °C (with an associated melting enthalpy of 94.34 J/g), whereas the major endothermic event in CAGE-IL was the sharp peak at 230.36 °C (with an associated melting enthalpy of 659.57 J/g) (Figure 1). Regarding the gel samples, the major endothermic event in the gel integrating pure CUR was the sharp peak at 92.33 °C (with an associated melting enthalpy of 832.37 J/g), and the major endothermic event in the gel integrating pure CUR and CAGE-IL was the sharp peak at 80.64 °C (with an associated melting enthalpy of 584.99 J/g). The effect of integrating CAGE-IL in the gel formulation was to substantially decrease the melting temperature of the gel formulation, with the gel formulation integrating pure CUR displaying a peak temperature displaced to lower temperatures, not far away from the melting point of water, since the gel formulations contained ca. 94% ultrapure water. As is apparent from inspection of the data obtained (Figure 1), the gel formulation integrating CAGE-IL and either pure CUR or commercial turmeric exhibited relatively high melting temperatures, allowing us to conclude that these gel formulations are stable for the intended application, since the temperature at the surface of the skin rarely exceeds 33.5 °C.

The animal model used to study psoriasis-like inflammation in mice with topical application of IMQ onto the depilated backs or ears of mice is the most popular model. This model has several advantages, the first one being that it is cheap and very easy to perform. Although applied topically, IMQ may induce weight loss and systemic cytokine elevation in mice, most likely due to ingestion (via licking) of the cream [57]. In the experiments performed, the animals were healthy, without noticeable problems related to their weight. The animal model used was thus considered adequate for the intended purpose. According to [57], 5% IMQ cream applied daily to the depilated backs or ears of mice for 5 to 7 days induces the development of an inflammatory response in mice similar to human psoriasis.

Both males and females were included in the animal trials with the aim of verifying whether there were differences in the induction of psoriasis by IMQ, as pointed out in [58]. In addition, the inclusion of female mice was thought to be interesting from the point of view of checking for changes in response due to estrous cycle variability [59].

It was observed that the gel formulations did not produce any signs of inflammation or peeling of the skin of the ears during the twenty days of application (male mice, Figure 2b,d,f; female mice, Figure 3b,d,f), thus showing very good compatibility with the epidermis of the skin of the animals’ ears. The application of the formulations in the respective groups also promoted no significant changes in the thickness of the ears in either male (Figure 4a) or female mice (Figure 4b), and the statistical analysis of the results obtained did not reveal any differences between the control groups (*p* > 0.05). In Figure 2, Figure 3, Figure 5 and Figure 6 one can observe that, in the groups of animals where the LBG gel with CUR and CAGE-IL was applied (photomicrographs *e* and *f* in Figure 2 and Figure 3, and photomicrographs *f* and *h* in Figure 5 and Figure 6), a yellowish coloration formed in the ears and in the nearby fur of the mice. This was probably a result of the greater solubilization of CUR by CAGE-IL, together with a greater transdermal permeation of CUR into the deeper layers of the skin. According to [60], the use of ionic liquids and deep eutectic solvents in biomedical applications has grown significantly due to their peculiar characteristics such as the ability to solubilize drugs.

Figure 7 shows the evolution of the severity of the inflammation of the epidermis (erythema) of the animals’ ears, induced by IMQ in male (Figure 7a) and female (Figure 7b) mice. After the 2nd day of IMQ application, the male mice’s ears showed a slight formation of erythema (mean value equal to 1), and on the 10th day of application they already had psoriasis (Figure 7a) (mean value equal to 3). A possible explanation might be a difference in response in male and female mice depending on the cyclic hormonal changes across the ovulatory cycle in females, introducing variability in the measures in comparison to males [59].

The erythema remained intense until the 16th day after the application of IMQ, remaining a mild manifestation until the 20th day of follow-up. The boxplot plot in Figure 7b shows the evolution of erythema in the ears of female mice after the application of IMQ. After the 2nd day of IMQ application, a slight formation of erythema was observed in the ears of female mice (mean value equal to 1) and on the 10th day of application the ears already presented psoriasis (mean value equal to 3). The erythema remained intense until the 16th day, and at the 20th day of follow-up the presence of erythema in the ears of female mice was not observed.

The boxplot plot in Figure 7c shows the evolution of epidermis desquamation for male mice after the application of IMQ. Peeling of the skin of the ears began on the 6th day after the application of IMQ, with observation of a mild to moderate peeling intensity (mean value equal to 1.5). On the 10th day after the application of IMQ, the peeling became intense (average value equal to 2.5). The desquamation of the skin of the ears of male mice (Figure 7c) continued to manifest itself until the 18th day after the application of IMQ (average value equal to 2.0), and on the 20th day, desquamation was still observed but with only slight intensity.

The boxplot plot in Figure 7d shows the evolution of epidermis desquamation in the female mouse ears after the application of IMQ. The peeling of the skin of the ears started on the 8th day after the application of IMQ, and a peeling of the skin of the ears of the female mice of light to moderate intensity (mean value equal to 1.5) was observed. On the 10th day after the application of IMQ, a slight increase in the PASI score was observed, with moderate intensity (mean value equal to 2.0). The desquamation of the skin of the ears of female mice appeared at slight intensity up to the 18th day after application of IMQ, and by the 20th day no more desquamation was observed.

By inspecting the median values of the boxplot graphs in Figure 8a,b, one can observe that the groups of animals exposed to IMQ showed erythema values close to 4, both in male mice (Figure 8a, *p* > 0.05) and in female mice (Figure 8b, *p* > 0.05), reaching the maximum lesion value according to the PASI scale. The LBG gel (2%, *w/w*) containing CUR (2%, *w/w*) and CAGE-IL (2%, *w/w*) was more effective compared to the other gel formulations (*p* < 0.001) in the treatment of erythema for both male (Figure 8a) and female (Figure 8b) mice.

The peeling of the skin caused by induction of psoriasis and its remission by the treatments (gel formulations) applied was also monitored, and the data recorded on the PASI scale. The boxplot graphs in Figure 8c,d show the results for the mean skin desquamation of the ears of male mice (Figure 8c) and female mice (Figure 8d) caused by the application of IMQ and the comparison of the remission obtained with treatment using the formulated gels. By inspecting the median values of the graphs, one can observe that the male mice belonging to the groups exposed to IMQ showed median values for skin desquamation of the ears ranging from 2 to 2.5 (Figure 8c), indicating moderate intensity of desquamation according to the PASI scale (*p* > 0.05), while the female mice belonging to the groups exposed to IMQ showed median values for peeling of the skin of the ears ranging from 1 to 2.0 (Figure 8d), also indicating moderate intensity of peeling according to the PASI scale (*p* > 0.05).

The LBG gel (2%, *w/w*) containing CUR (2%, *w/w*) and CAGE-IL (2%, *w/w*) was more effective than the other gel formulations (*p* < 0.001) in the treatment of desquamation of the skin of the ears for both male mice (Figure 8c) and female mice (Figure 8d).

Thus, according to the results obtained in this research, the LBG gel (2%, *w/w*) containing CUR (2%, *w/w*) and CAGE-IL (2%, *w/w*) showed high potential for the treatment of psoriasis, as it promoted the remission of both erythema and scaling of the skin produced in the animal model of induction of psoriasis with IMQ.

In the absolute control group (G1, male mice, Figure 2Æ-m; G1 female mice, Figure 3Æ-f), the granular layer was present and there were no nuclei in the stratum corneum (thin arrow). Cornified cells in the section had a loose reticulated appearance typical of orthokeratosis (thick arrow). In group G2, after 20 d of application of LBG gel (2%, *w/w*) in male mice ears (Figure 2b1) and in female mice ears (Figure 3b1), there were no changes in the granular layer, which was present, and there were no nuclei in the stratum corneum (thin arrow). Cornified cells in section had a loose reticulated appearance typical of orthokeratosis (thick arrow). In group G3, after 20 d of application of LBG gel (2%, *w/w*) containing CUR (2%, *w/w*) in male mice ears (Figure 2d1) and in female mice ears (Figure 3d1), there were no changes in the granular layer, which was present, and there were no nuclei in the stratum corneum (thin arrow). Cornified cells in section had a loose reticulated appearance typical of orthokeratosis (thick arrow). In group G4, after 20 d of application of LBG gel (2%, *w/w*) containing CUR (2%, *w/w*) and CAGE-IL (2%, *w/w*) in male mice ears (Figure 2f1) and in female mice ears (Figure 3f1), there were no changes in the granular layer, which was present, and there were no nuclei in the stratum corneum (thin arrow). Cornified cells in section had a loose reticulated appearance typical of orthokeratosis (thick arrow).

In the IMQ control group (G5, male mice, Figure 5b1; G5 female mice, Figure 6b1), the presence of considerable thickening of the epidermis due to an increase in the cells of the spinous layer (acanthosis) was observed (thick arrow). The maturation of epidermal cells was accelerated, with parakeratosis being noted, due to the presence of nuclei in the stratum corneum and the absence of the granular layer, and therefore the absence of keratohyalin granules (thin arrow). Changes had occurred in the granular layer, which was not present, and there were nuclei in the stratum corneum (arrowhead). In group G6, after 10 d of application of LBG gel (2%, *w/w*) in male mice ears (Figure 5d1) and in female mice ears (Figure 6d1), application of the LBG gel (2%, *w/w*) had slightly decreased acanthosis (thick arrow), as well as parakeratosis; the presence of nuclei in the stratum corneum and an absence of the granular layer, and therefore an absence of keratohyalin granules (thin arrow), were also observed. Changes had occurred in the granular layer, which was not present, and there were nuclei in the stratum corneum (arrowhead). In group G7, after 10 d of application of LBG gel (2%, *w/w*) containing CUR (2%, *w/w*) in male mice ears (Figure 5f1) and in female mice ears (Figure 6f1), application of the LBG gel (2%, *w/w*) containing CUR (2%, *w/w*) promoted a reduction in acanthosis (thick arrow), with a reduction in parakeratosis and the presence of few nuclei in the stratum corneum (thin arrow). Cornified cells returned to a loose reticulated appearance (arrowhead). In group G8, after 10 d of application of LBG gel (2%, *w/w*) containing CUR (2%, *w/w*) and CAGE-IL (2%, *w/w*) in male mice ears (Figure 5h1) and in female mice ears (Figure 6h1), application of the LBG gel (2%, *w/w*) containing CUR (2%, *w/w*) and CAGE-IL (2%, *w/w*) promoted considerable reduction of acanthosis (thick arrow). There was a reduction in parakeratosis, and few nuclei were present in the stratum corneum (thin arrow). Cornified cells in section again showed a loose reticulated appearance, indicating the formation of orthokeratosis (arrowhead). The LBG gel (2%, *w/w*) containing CUR (2%, *w/w*) and CAGE-IL (2%, *w/w*) thus showed high potential for the treatment of psoriasis, reversing the histological manifestations of psoriasis to a state very close to normal.

By inspecting the results shown in Figure 9, it can be seen that after 20 d of application of the formulated gels, there was no change in the thickness of the epidermis of the ears of the mice, in either males or females, revealing that the gels are safe to use. The one-way ANOVA statistical analysis performed on the experimental data showed that there were no differences between the groups where gels were applied and the absolute control group (*p* > 0.05), for either male mice (Figure 9a) or female mice (Figure 9b). The groups of animals subjected to the application of IMQ showed a considerable increase in the thickness of the epidermis of the ears, observed in both male mice and female mice, with no statistical differences (*p* > 0.05) between them. Treatment with LBG gel (2%, *w/w*) had the same effect as the natural response of the mice organisms, irrespective of gender. Compared with the other treatments, treatment for 10 consecutive days with LBG gel (2%, *w/w*) containing CUR (2%, *w/w*) and CAGE-IL (2%, *w/w*) significantly reduced the thickness of the epidermis of the ears in both male (Figure 9a) and female (Figure 9b) mice compared to the untreated psoriasis group (*p* < 0.001). Thus, treatment for 10 consecutive days with LBG gel (2%, *w/w*) containing CUR (2%, *w/w*) and CAGE-IL (2%, *w/w*) was shown to produce a much better remission of the thickness of the epidermis of the skin of the ears of the mice, irrespective of gender, with results very close to those of the absolute control group.

In previous studies by our research group [19,33] the LBG gel formulation containing CUR at 2% (*w/w*) and CAGE-IL at 2% (*w/w*) promoted the best results in terms of the amount of CUR retained within the skin matrix. This result is highly significant if this gel formulation is intended to be used in the topical treatment of psoriasis, since at 2% (*w/w*), CAGE-IL allows the highest level of transdermal permeation of CUR [19,33] while allowing saturation of the skin with the highest amount of CUR.

According to Boscariol et al. [19,33], CAGE-IL has, in general, the ability to increase transdermal permeation of biomacromolecules, using mechanisms such as transient disruption of the intercellular and intracellular lipids and protein organization of the stratum corneum, fluidization and the creation of diffusional pathways [19].

It is known that psoriasis is a chronic, complex and multifactorial inflammatory autoimmune skin disease, with affected individuals experiencing significant social and psychological problems with a negative impact on their quality of life. It affects both men and women around the world [61], with women being more impacted, experiencing fluctuations in disease activity due to hormonal changes and with the stages of a journey towards motherhood [62,63]. These aspects of the manifestations of psoriasis motivated the research described herein, where the induction and manifestation of psoriasis was studied in an animal model with male and female mice. The initial statistical analysis reported in this research showed that both males and females responded to the psoriasis induction procedures, and as early as the third day, the female mice of groups G5 (*p* = 0.0470), G6 (*p* = 0.0030), G7 (*p* = 0.0090) and G8 (*p* = 0.0090) treated with IMQ already presented erythema, whereas in the male mice, the manifestation of erythema was observed on the fourth day in groups G5 (*p* = 0.0002), G6 (*p* = 0.0002), G7 (*p* = 0.0001) and G8 (*p* = 0.0001). Regarding the treatment with the formulated gels, the male mice, on the fourth day, in groups G7 (*p* = 0.9770) and G8 (*p* = 0.9770), showed improvement in relation to the control group G5 for desquamation, whereas the female mice in groups G7 (*p* = 0.9830) and G8 (*p* = 0.9820) also presented similar results compared to the control group G5. Ear thickness reduction results were observed in male mice in groups G7 (*p* = 0.0860) and G8 (*p* = 0.0867), but the thickness did not return to normal values compared to group G1. Female mice showed a reduction in the thickness of the ears to the normal range in groups G7 (*p* = 0.2200) and G8 (*p* = 0.2230) compared to group G1.

A second statistical analysis was performed: the Tukey HSD test with unequal n (a generalization of Tukey’s test to the case of unequal samples sizes), without separating male and female mice, and considering each group of 5 mice, with n = 10 for the ears. The statistical analysis revealed that after 10 days of induction of psoriasis by IMQ, groups G5, G6, G7 and G8 (all with *p*-value = 0.00123) already presented all the manifestations of psoriasis, and that for the formulated gels, groups G2, G3 and G4 did not develop any signs of psoriasis compared to group G1 (*p* = 1.0000), proving that the gel formulations were non-toxic. On the second day of treatment, groups G7 (*p* = 0.0297) and G8 (*p* = 0.000149) both showed improved erythema compared with group G5. Group G6 (*p* = 0.0193) showed improved psoriasis erythema only on the eighth day of treatment. On the fourth day after initiating treatment, group G8 (*p* = 0.0924) showed a reduction in desquamation compared to the control, returning to normality.

In groups G2 (*p* = 0.9370), G3 (*p* = 0.9900) and G4 (*p* = 1.000) no changes were observed in ear thickness. On the second day of treatment, group G8 (*p* = 0.000123) already showed an improvement in ear thickness, with a significant reduction compared to group G5. On the fourth day, group G7 (*p* = 0.000126) also showed a decrease in ear thickness. Group G6 (*p* = 1.000) did not show an improvement in ear thickness, even after 10 days of treatment.

Finally, it was observed that performing a statistical analysis after randomly dividing male and female mice between the groups, with a sample of 2 mice (n = 4 ears) or 3 mice (n = 6 ears), produced results that were comparable to analyzing a sample of 5 mice (n = 10 ears). The division of the mice into males and females was important for the study of possible variations in the manifestation of psoriasis between genders.

## 5. Conclusions

Approximately 80% of patients with mild to moderate psoriasis can be treated with topical therapies. The advantages of topical treatments over traditional systemic treatments include availability, lower costs and lack of safety issues. Furthermore, topical therapies can empower patients to take control of the psoriatic disease. However, to increase the effectiveness of bioactive molecules, it is often necessary to increase the skin permeability to the biomolecules. Curcumin is a good example. Ionic liquids, in particular choline and geranic acid ionic liquid (CAGE-IL), have shown great promise in drug delivery applications and have been used to enhance the transdermal delivery of several molecules. CAGE-IL has the potential to become a powerful excipient to improve the pharmacokinetic profiles of many pharmaceutically active molecules in transdermal applications. The effect of an LBG gel formulation containing curcumin at 2% (*w/w*) and CAGE-IL at 2% (*w/w*) in controlling psoriasis was studied in vivo using a mouse model with psoriasis induced by imiquimod. The LBG gel (2%, *w/w*) containing curcumin (2%, *w/w*) and CAGE-IL (2%, *w/w*) showed a high potential for applications in the treatment of psoriasis, reversing the histological manifestations of psoriasis to a state very close to normal, with clear benefits to patients suffering from psoriasis. From the results obtained in this study, it can be said that the association between curcumin and CAGE-IL in pharmaceutical formulations is successful for the transdermal delivery of curcumin in the topical treatment of psoriasis.

## Figures and Tables

**Figure 1 pharmaceutics-14-00779-f001:**
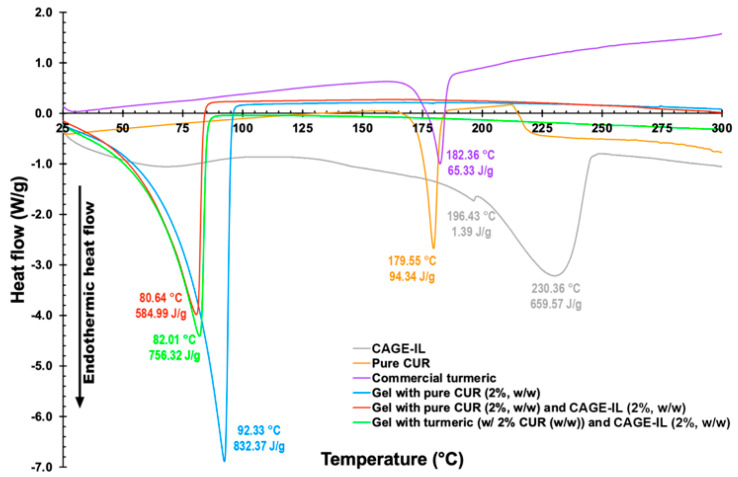
Differential scanning calorimetry thermograms of CAGE-IL (grey line), pure CUR (orange line), commercial turmeric (purple line), LBG gel formulation integrating pure CUR (blue line), LBG gel formulation integrating pure CUR and CAGE-IL (red line) and LBG gel formulation integrating commercial turmeric and CAGE-IL (green line).

**Figure 2 pharmaceutics-14-00779-f002:**
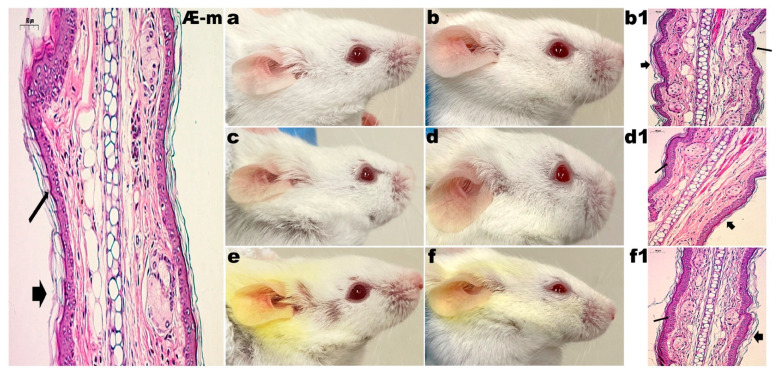
Results obtained after application of the formulations in the ears of male mice. (**a**) (G2) 10 days of LBG (2%, *w/w*) gel application; (**b**) (G2) 20 days of LBG (2%, *w/w*) gel application; (**c**) (G3) 10 days of application of LBG gel (2%, *w/w*) with CUR (2%, *w/w*); (**d**) (G3) 20 days of application of LBG gel (2%, *w/w*) with CUR (2%, *w/w*); (**e**) (G4) 10 days of application of LBG gel (2%, *w/w*) with CUR (2%, *w/w*) and CAGE-IL (2%, *w/w*); (**f**) (G4) 20 days of application of LBG gel (2%, *w/w*) with CUR (2%, *w/w*) and CAGE-IL (2%, *w/w*); (**Æ**-**m**) (G1) photomicrograph (x200 magnification) of histological section of mouse ear skin of male absolute control; (**b1**) (G2) photomicrograph (×200 magnification) of histological section of mouse ear skin after 20 d of application of LBG gel (2%, *w/w*); (**d1**) (G3) photomicrograph (×200 magnification) of histological section of mouse ear skin after 20 d of application of LBG gel (2%, *w/w*) with CUR (2%, *w/w*); (**f1**) (G4) photomicrograph (×200 magnification) of histological section of mouse ear skin after 20 d of application of LBG gel (2%, *w/w*) with CUR (2%, *w/w*) and CAGE-IL (2%, *w/w*). The histological sections were stained with hematoxylin and eosin. For the significance of the inserted (thick, thin and arrowhead) arrows, please refer to the Discussion section. Scale bar represents 50 µm. Note: the yellowish color observed in the fur of the animals is due to the use of the gel with CUR and CAGE-IL, with the ionic liquid promoting a greater solubility of CUR and consequently dying the fur.

**Figure 3 pharmaceutics-14-00779-f003:**
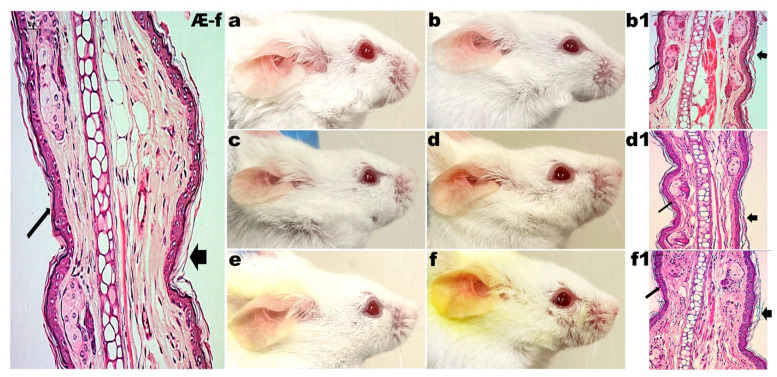
Results obtained after application of the formulations in the ears of female mice. (**a**) (G2) 10 days of LBG (2%, *w/w*) gel application; (**b**) (G2) 20 days of LBG (2%, *w/w*) gel application; (**c**) (G3) 10 days of application of LBG gel (2%, *w/w*) with CUR (2%, *w/w*); (**d**) (G3) 20 days of application of LBG gel (2%, *w/w*) with CUR (2%, *w/w*); (**e**) (G4) 10 days of application of LBG gel (2%, *w/w*) with CUR (2%, *w/w*) and CAGE-IL (2%, *w/w*); (**f**) (G4) 20 days of application of LBG gel (2%, *w/w*) with CUR (2%, *w/w*) and CAGE-IL (2%, *w/w*); (**Æ**-**f**) (G1) photomicrograph (×200 magnification) of histological section of mouse ear skin of female absolute control; (**b1**) (G2) photomicrograph (×200 magnification) of histological section of mouse ear skin after 20 d of application of LBG gel (2%, *w/w*); (**d1**) (G3) photomicrograph (×200 magnification) of histological section of mouse ear skin after 20 d of application of LBG gel (2%, *w/w*) with CUR (2%, *w/w*); (**f1**) (G4) photomicrograph (×200 magnification) of histological section of mouse ear skin after 20 d of application of LBG gel (2%, *w/w*) with CUR (2%, *w/w*) and CAGE-IL (2%, *w/w*). The histological sections were stained with hematoxylin and eosin. For the significance of inserted (thick, thin and arrowhead) arrows, please refer to the Discussion section. Scale bar represents 50 µm. Note: the yellowish color observed in the fur of the animals is due to the use of the gel with CUR and CAGE-IL, with the ionic liquid promoting a greater solubility of CUR and consequently dying the fur.

**Figure 4 pharmaceutics-14-00779-f004:**
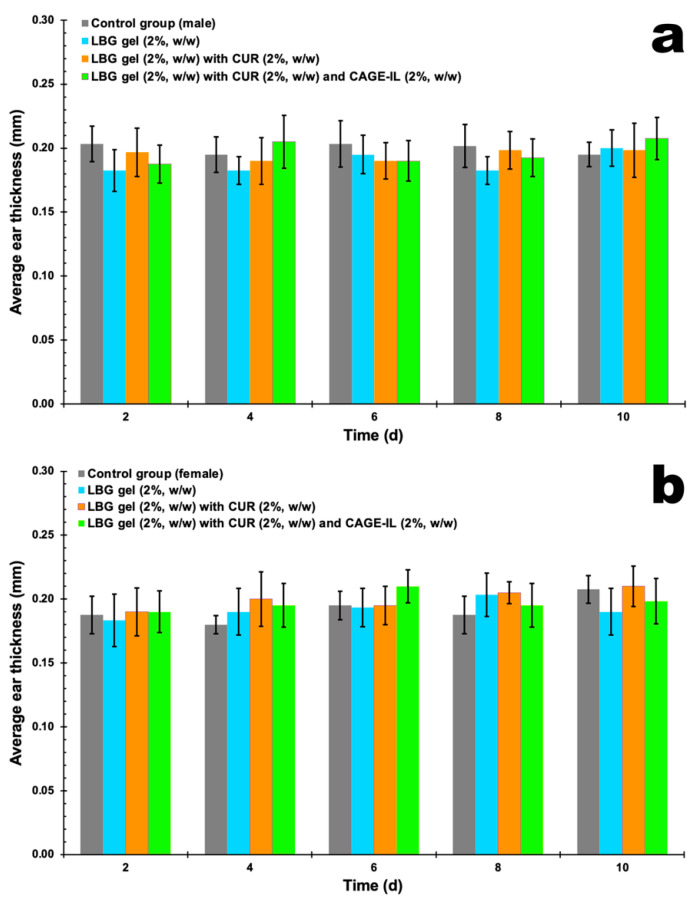
Results obtained for the evolution of the thickness of the ears of the animals in the control groups, during 20 d of application of LBG gel (2%, *w/w*), LBG gel (2%, *w/w*) with CUR (2%, *w/w*) and LBG gel (2%, *w/w*) with CUR (2%, *w/w*) and CAGE-IL (2%, *w/w*). (**a**) Male mice (*p* > 0.05) and (**b**) female mice (*p* > 0.05). The results displayed are the average of three determinations, and the error bars represent the standard deviations.

**Figure 5 pharmaceutics-14-00779-f005:**
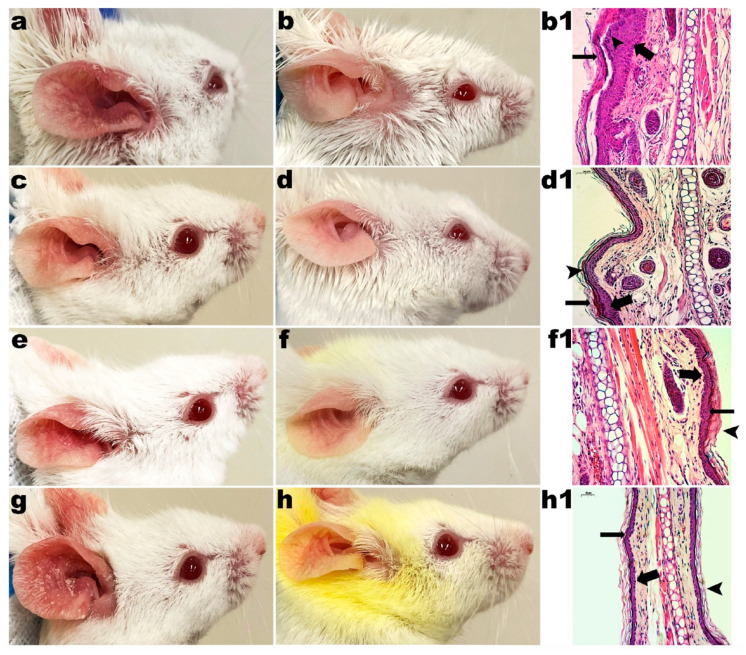
Results obtained after IMQ-induced psoriasis and application of the formulations in the ears of male mice. (**a**) (G5) 10 days of application of IMQ; (**b**) (G5) natural recovery of the mouse ears without treatment; (**c**) (G6) 10 days of application of IMQ; (**d**) (G6) 10 days of application of LBG (2%, *w/w*) gel; (**e**) (G7) 10 days of application of IMQ; (**f**) (G7) 10 days of application of LBG gel (2%, *w/w*) with CUR (2%, *w/w*); (**g**) (G8) 10 days of application of IMQ; (**h**) (G8) 10 days of application of LBG gel (2%, *w/w*) with CUR (2%, *w/w*) and CAGE-IL (2%, *w/w*); (**b1**) (G5) photomicrograph (×200 magnification) of histological section of mouse ear skin after 20 d of natural recovery of IMQ-induced psoriasis; (**d1**) (G6) photomicrograph (×200 magnification) of histological section of mouse ear skin with IMQ-induced psoriasis after 20 d of application of LBG gel (2%, *w/w*); (**f1**) (G7) photomicrograph (×200 magnification) of histological section of mouse ear skin with IMQ-induced psoriasis after 20 d of application of LBG gel (2%, *w/w*) with CUR (2%, *w/w*); (**h1**) (G8) photomicrograph (×200 magnification) of histological section of mouse ear skin with IMQ-induced psoriasis after 20 d of application of LBG gel (2%, *w/w*) with CUR (2%, *w/w*) and CAGE-IL (2%, *w/w*). All histological sections were stained with hematoxylin and eosin. For the significance of inserted (thick, thin and arrowhead) arrows, please refer to the Discussion section. Scale bar represents 50 µm. Note: the yellowish color observed in the fur of the animals is due to the use of the gel with CUR and CAGE-IL, with the ionic liquid promoting a greater solubility of CUR and consequently dying the fur.

**Figure 6 pharmaceutics-14-00779-f006:**
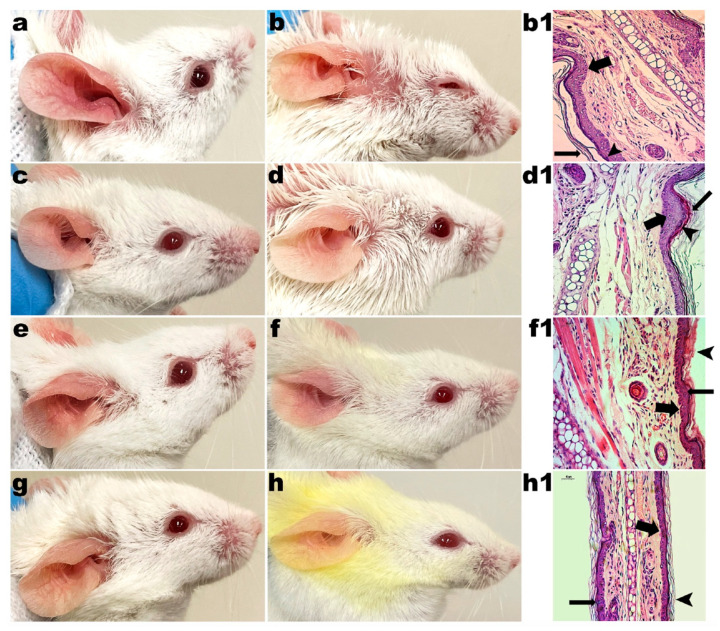
Results obtained after IMQ-induced psoriasis and application of the formulations in the ears of female mice. (**a**) (G5) 10 days of application of IMQ; (**b**) (G5) natural recovery of the mouse ears without treatment; (**c**) (G6) 10 days of application of IMQ; (**d**) (G6) 10 days of application of LBG (2%, *w/w*) gel; (**e**) (G7) 10 days of application of IMQ; (**f**) (G7) 10 days of application of LBG gel (2%, *w/w*) with CUR (2%, *w/w*); (**g**) (G8) 10 days of application of IMQ; (**h**) (G8) 10 days of application of LBG gel (2%, *w/w*) with CUR (2%, *w/w*) and CAGE-IL (2%, *w/w*); (**b1**) (G5) photomicrograph (×200 magnification) of histological section of mouse ear skin after 20 d of natural recovery of IMQ-induced psoriasis; (**d1**) (G6) photomicrograph (x200 magnification) of histological section of mouse ear skin with IMQ-induced psoriasis after 20 d of application of LBG gel (2%, *w/w*); (**f1**) (G7) photomicrograph (×200 magnification) of histological section of mouse ear skin with IMQ-induced psoriasis after 20 d of application of LBG gel (2%, *w/w*) with CUR (2%, *w/w*); (**h1**) (G8) photomicrograph (×200 magnification) of histological section of mouse ear skin with IMQ-induced psoriasis after 20 d of application of LBG gel (2%, *w/w*) with CUR (2%, *w/w*) and CAGE-IL (2%, *w/w*). All histological sections were stained with hematoxylin and eosin. For the significance of inserted (thick, thin and arrowhead) arrows, please refer to the Discussion section. Scale bar represents 50 µm. Note: the yellowish color observed in the fur of the animals is due to the use of the gel with CUR and CAGE-IL, with the ionic liquid promoting a greater solubility of CUR and consequently dying the fur.

**Figure 7 pharmaceutics-14-00779-f007:**
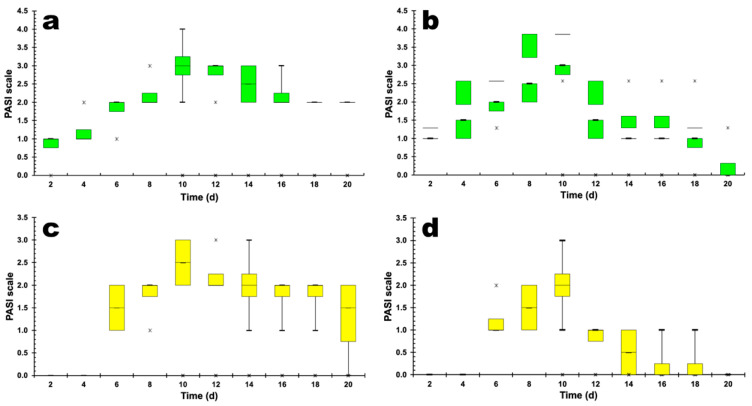
Boxplot graphs of the evolution of erythema (**a**,**b**) and ear skin peeling (**c**,**d**) in male (**a**,**c**) (*p* < 0.001) and female (**b**,**d**) (*p* < 0.001) mice. Legend: (🞵) represent outlier points off the average values.

**Figure 8 pharmaceutics-14-00779-f008:**
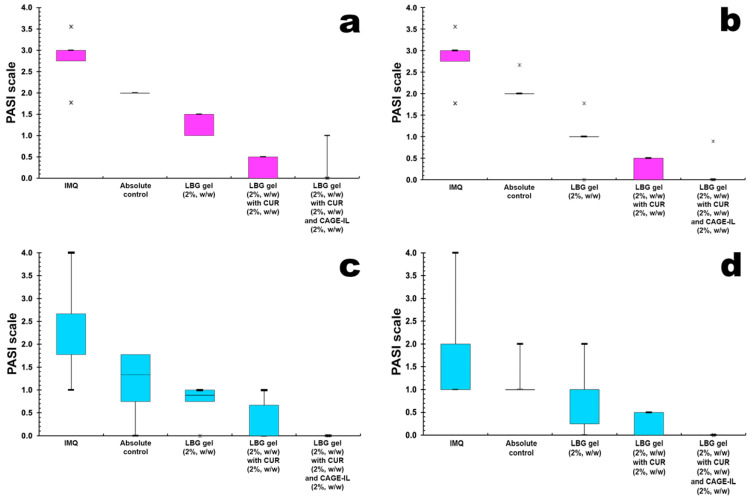
Boxplot graphs of the evolution of erythema (**a**,**b**) and ear skin peeling (**c**,**d**) induced by imiquimod (IMQ) and the comparison of the remission obtained via application of the treatments with the LBG gel formulations in male (**a**,**c**) (*p* < 0.001) and female (**b**,**d**) (*p* < 0.001) mice. Legend: (🞵) represent outlier points off the average values.

**Figure 9 pharmaceutics-14-00779-f009:**
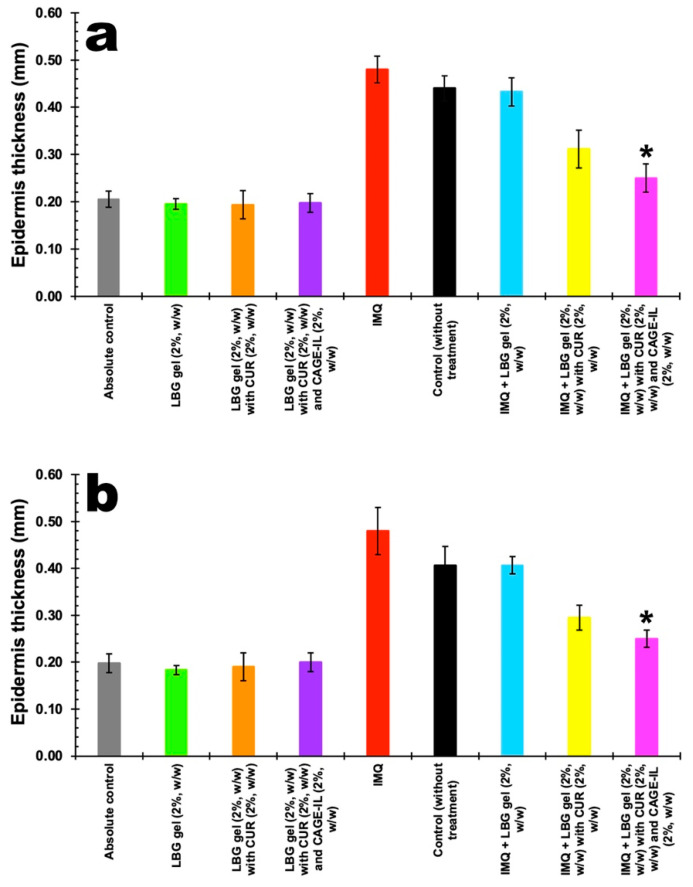
Results obtained for the thickness of the epidermis of the ears of the animals in the control and treatment groups, after 20 d of application of LBG gel (2%, *w/w*), LBG gel (2%, *w/w*) with CUR (2%, *w/w*) and LBG gel (2%, *w/w*) with CUR (2%, *w/w*) and CAGE-IL (2%, *w/w*), and after 10 d following induction of psoriasis with IMQ. (**a**) Male mice (*p* > 0.05) and (**b**) female mice (*p* > 0.05). The results displayed are the averages of three determinations and the error bars represent the standard deviations. The data were statistically analyzed via one-way ANOVA, and ***** represents a statistically significant result compared with the control group (without treatment, black columns).

**Table 1 pharmaceutics-14-00779-t001:** Detailed composition of all gel formulations prepared.

Compound	Function in the Formulation	Formulation					
		{LBG} Gel	{LBG + CUR} Gel	{LBG + CUR + CAGE − IL} Gel	{LBG} Gel	{LBG + CUR} Gel	{LBG+ CUR + CAGE − IL} Gel
		% (*w/w*)			Mass (g)		
Locust bean gum (LBG)	Gel-forming agent	2	2	2	2.000	2.000	2.000
Curcumin (CUR)	Bioactive compound	-----	2	2	2.000	2.000	2.000
Choline and geranic acid ionic liquid (CAGE-IL)	Skin permeation enhancer	-----	-----	2	-----	-----	2.000
Methylparaben	Preservative, antifungal	0.1	0.1	0.1	0.100	0.100	0.100
Ultrapure water	Solvent	up to 100	up to 100	up to 100	up to 100	up to 100	up to 100

LBG gel: gel formulation with locust bean gum; {LBG + CUR} gel: gel formulation with locust bean gum and curcumin; {LBG + CUR + CAGE − IL} gel: gel formulation with locust bean gum, curcumin and choline and geranic acid ionic liquid.

## Data Availability

Data will be made available to researchers upon request.

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
