# Peer review of "Transdermal Permeation Assays of Curcumin Aided by CAGE-IL: In Vivo Control of Psoriasis"

_pharmaceutics, 2022, doi:10.3390/pharmaceutics14040779_

Round 1
Reviewer 1 Report
The authors have conducted a study on psoriasis, a skin disease that has no cure so far, by synthesizing a bio-polysaccharide hydrogel formulation that combines curcumin and ionic liquid (CAGE-IL) as a percutaneous skin penetration promoter. The manuscript showed the possibility that gel development containing curcumin and liquid ion (CAGE-IL) could be applied to psoriasis treatment, reversing the histological signs of psoriasis to a very close state. The explanation of the results was easy to see and understand, but I would like to ask you some questions about the manuscript.
Major
- The biopolysaccharide gel was synthesized using curcumin and CAGE-IL, and there is no information about the formulation of the gel, and the evaluation of the characterization is only data on the differential scanning calorimetry (DSC). It would be great if you could evaluate more diverse characterizations.
- We treated the symptoms mainly in the local area (ear), but are there any other problems or precautions when applying this treatment to the entire area?
- The reason why Choline and geranic acid were selected in the 1. Introduction part is not clearly revealed, so what are the reasons for selecting them among ionic liquids and the advantages of each substance?
- In Fig 5 and 6, is there a reason we have seen imiquimod-induced psoriasis in the ears of mice for 20 days? Can you do more than that? Can a longer period of drug administration return to normal rat ear thickness?
- Overall, in almost figures, the notion of statistical significance is hard for the readers to understand. Especially, in figure 9, although LBG gel containing curcumin (2%, w/w) and CAGE-IL (2%, w/w) group showed a significant decreases in the thickness of the epidermal of the ear compared to the untreated psoriasis group, it is not shown on the graph.
- In figure 7, the remission of erythema and ear skin peeling between the genders was different from the 12 days. The remission rate tends to be high in females, please explain the cause.
- Why did you set the relative humidity at 60-70% in the mouse experiment? Psoriasis gets worse in dry environments, and I am curious about the treatment effect in humid environments.
- Are you sure psoriasis was caused by IMQ cream only in 10 days? PASI is subjective evaluation. Did you perform other indicators to evaluate the severity of psoriasis?
- Why the presence or absence of CAGE-IL affects the coloration of curcumin-yellow?
- In Figure 9, how did you measure the epidermal thickness? How do you know if by the symptoms of psoriasis or by swollenness?
- Figure 9. Usually, with treatment, it returns to its original state. Here, after treatment (IMQ+LBG gel with CUR and CAGE-IL), are there any experiments that have been observed after 20 days until the epidermis thickness returns to control?
Minor
- Thank you for filling out the processed drug for feature, which conducted a histological analysis of the ear, but if you write down the group of the model, it will be easier to understand.
- I would appreciate it if you could add scale bar to the photo where we performed your histological analysis.
- I would like you to add the p value for the overall graph. It seems necessary to distinguish whether the data is unreliable.
- 1, 2.2 and 2.2.2 The font is different. I think you intended it, but if you intended it, it would be more readable if you could distinguish it from other writings.
- 2.2. When making Biopolysaccharide gel with curcumin, the role of using locust bean gum is not shown, so why do you use it?
- A further explanation is needed for Figure 1. It is not easy to understand the meaning of the graph at once, so it would be good to understand if there is an additional explanation
- In Figures 2, 3, 5, and 6, the figure showing the skin tissue chart is also not bad to show before and after drug treatment.
- P16, line 526, “female 1 mice” should be “female mice”.
- P16, line 532, line 535, and P17, line 543, “2.0%” should be “2%”.
- It's hard to understand what it means to be yellow in a mouse figure (2,3,5,6). It is necessary to mention this.
- CAGE-IL is defined as “curcumin and ionic liquid” in 15th and 17th line, and as “Choline and geranic acid ionic liquid” in 75th
- LBG was not defined in the first use.
- Why the presence or absence of CAGE-IL affects the coloration of curcumin-yellow?
- In the description of the figure 2,3,5,6, There are some “gel LBG”.
3.2. Macroscopic/microscopic analysis of mice ears with about 20 trials, this study showed that it was effective in treating psoriasis. Then, after that, have you ever thought about how to speed up the treatment period?
Reviewer 2 Report
Journal: MDPI - Pharmaceutics
Manuscript
Title: Review "Transdermal permeation assays of curcumin aided by CAGE-2 IL: in vivo control of psoriasis"
Author(s): Rodrigo Boscariol, Erika A. Caetano, Denise Grotto, Raquel M. Rosa-Castro, José M. Oliveira Jr., Marta M. D. C. Vila and Victor M. Balcao
Reviewer Comments to Author(s)
Recommendation: Major revisions
In this article the author(s) have focused and outlined the transdermal application of curcumin by CAGE-IL containing gels. This is a comprehensive and well-written research work. However, there are some comments/concerns that could help and further develop the article.
The author(s) might consider the following:
- In the Materials and Methods: detailed concentrations should be provided for the synthesis of the gel in terms of clarity and reproducibility not only weight percentages, or references of detailed synthetic procedure if has already been published. Please state all the abbreviations before their use for example in line 132 LBG probably refers to locust bean gum.
- In the group division there is a total of 5 mice in each group, however each group has a different number of male and female mice and I the results for example figure 4 the average of male and female mice is presented. In many cases this average refers to only 2 mice. How this could affect the statistical significance of your results? Why the authors divided the female and male mice in such a way and in so many controls resulting in using a great number of animals and having only 2 males and only 2 females in each group with IMQ? How accurate the statistical analysis in figure 9 is for the groups of only 2 males or females?
- It is not quite clear the reason why the authors used female and male mice for the in vivo psoriasis study. Male and female patients have equal rates of psoriasis occurrence or are there any characteristic differences? Moreover, did the authors expected characteristic differences for male and female mice?
- Have the authors studied here or elsewhere the penetration level of curcumin from the CAGE-IL containing gel used in this study?
Reviewer 3 Report
This manuscript describes the potential of choline and geranic acid ionic liquid containing curcumin for topical therapy in psoriasis. The authors continued their interest in this field and extended the applications of ionic gel for the effective therapy of curcumin by skin delivery. The study was thoroughly designed and the data are presented in a good way. The outcome of this study will provide more scope in the future. I have a few minor comments/suggestions to improve the overall quality of this manuscript.
Comments
- Please mention how the composition of each formulation is selected. Whether the authors used anyway to optimize the formulations?
- Why Nitrogen was included in ionic gel?
- The only characterization method mentioned for the formulation is DSC. How about the other physicochemical properties like viscosity, spreadability, SEM/TEM, etc.?
- The DSC results are not described; what difference do the authors observe between various formulations?
- How about the systemic exposure of these formulations?
- The clinical relevance of this study needs to be included in the conclusion section. Also, authors may add a statement regarding extending the application of this product in other topical therapies including, skin, other mucoadhesive, and bioadhesive systems like ocular, nasal, buccal, vaginal, rectal, and oral drug delivery.
Round 2
Reviewer 1 Report
The authors have addressed all the comments and the revised manuscript may be considered for publication. I recommend "accept".
Reviewer 2 Report
Journal: MDPI - Pharmaceutics
Manuscript
Title: Review "Transdermal permeation assays of curcumin aided by CAGE-2 IL: in vivo control of psoriasis"
Author(s): Rodrigo Boscariol, Erika A. Caetano, Denise Grotto, Raquel M. Rosa-Castro, José M. Oliveira Jr., Marta M. D. C. Vila and Victor M. Balcao
Reviewer Comments to Author(s)
Recommendation: Accept
In this article the author(s) have made a comprehensive and interesting research work for transdermal application of curcumin. The correction and clarifications are adequate and the article can be accepted for publication.
